# Australian Consumers’ Attitudes towards Sustainable Diet Practices Regarding Food Waste, Food Processing, and the Health Aspects of Diet: A Cross Sectional Survey

**DOI:** 10.3390/ijerph20032633

**Published:** 2023-02-01

**Authors:** Janelle D. Healy, Satvinder S. Dhaliwal, Christina M. Pollard, Piyush Sharma, Clare Whitton, Lauren C. Blekkenhorst, Carol J. Boushey, Jane A. Scott, Deborah A. Kerr

**Affiliations:** 1School of Population Health, Curtin University, Kent Street, P.O. Box U1987, Perth, WA 6845, Australia; 2Curtin Health Innovation Research Institute, Curtin University, Kent Street, P.O. Box U1987, Perth, WA 6845, Australia; 3Duke-NUS Medical School, National University of Singapore, College Road, Singapore 169857, Singapore; 4Institute for Research in Molecular Medicine (INFORMM), University Sains Malaysia, Minden 11800, Pulau Pinang, Malaysia; 5Singapore University of Social Sciences, 463 Clementi Road, Singapore 599494, Singapore; 6Enable Institute, Faculty of Health Sciences, Curtin University, Bentley, WA 6102, Australia; 7School of Management and Marketing, Curtin University, Kent Street, P.O. Box U1987, Perth, WA 6845, Australia; 8Nutrition and Health Innovation Research Institute, School of Medical and Health Sciences, Royal Perth Hospital Research Foundation, Edith Cowan University, Perth, WA 6000, Australia; 9Epidemiology Program, University of Hawaii Cancer Center, Honolulu, HI 96813, USA

**Keywords:** sustainable diets, food waste, environmental sustainability, food processing, dietary guidelines

## Abstract

Environmentally sustainable diets are increasingly aspired to in food-based dietary guidelines across the world. However, little is known about consumer attitudes toward these diets when making food decisions. This study aimed to identify the demographic characteristics of Australian adults based on the level of attention they paid to the healthfulness of their diet, their consideration of the level of food processing, and their concern about household food waste and sustainable packaging disposal. Adults aged from 18 to over 75 years (n = 540) were surveyed online. Thirty-seven percent were concerned about sustainable food waste, 28% considered the level of food processing when making food decisions, and 23% paid attention to the healthfulness of the food they ate. Adults who had higher educational attainment (above Year 12) were twice as likely to be concerned about food waste and sustainable packaging disposal (odds ratio (OR) = 2.10, 95% confidence interval (CI) 1.29–3.4), and processing levels (OR = 2.04, 95% CI 1.23–3.42) (controlling for age and gender). Those earning an income over AUD$100,000 were twice as likely to pay attention to the healthfulness of their food choices than those earning less than AUD$50,000 (OR = 2.19, 95% CI 1.28–3.74). Only 9% percent were concerned about or paid attention to all three of the components of healthy sustainable diets investigated, and 45% paid no attention and were not concerned about all three components. These findings suggest there is a need to educate the public to raise awareness of and concern for healthy, minimally processed, and sustainable food choices.

## 1. Introduction

Sustainable healthy diets are defined as “dietary patterns that promote all dimensions of individuals’ health and wellbeing; have low environmental pressure and impact; are accessible, affordable, safe and equitable; and are culturally acceptable”. [1] (p. 8), are increasingly aspired to in dietary guidelines. Environmental considerations build on biological sciences to understand the ecological interplay between animals, plants, and the environment [2]. Authoritative bodies and emerging research recommend sustainable eating for the best planetary and human health outcomes [1,2,3]. 

Dietary guidelines (DGs) in some countries, such as Brazil and Israel, provide advice to consumers on the adverse health effects of heavily processed foods and concurrently encourage environmentally sustainable food choices [4]. The DGs provide evidence-based advice on dietary patterns for the health of both the individual and the population [5]. The Australian Dietary Guidelines (ADGs) do not include specific guidelines about sustainable eating patterns as the agreement was not reached on this guidance even following additional consultations. The information regarding environmental sustainability was incorporated into Appendix G, entitled ‘Food, nutrition and environmental sustainability’ [6] (p. 130). Most Australians do not consume diets that meet the ADGs [7], and consequently, many are at risk of, or already have, avoidable non-communicable diet-related diseases [7]. Australian health surveys consistently highlight two concurrent problems. Firstly, there is an inadequate intake of nutritious healthy foods; secondly, there is an excess intake of unhealthy food (called ‘discretionary’ foods in ADGs) which are high in added fat, sugar, and salt and are usually highly processed [7]. 

There is little information available about Australian adults’ attitudes toward healthy and sustainable food choices. Harray et al. (2017) explored young adults’ dietary perceptions and intakes and found that those who were concerned about the health aspects of their diet consumed less discretionary food [8]. This then led to the development of a healthy and sustainable diet index (HSDI) [9] to measure adherence to a sustainable diet across five categories related to environmental sustainability, including ultra-processed energy-dense nutrient-poor foods, packaged foods, and food waste [9]. The lower the attention paid to the health aspects of diet, the poorer the dietary quality and environmentally sustainable eating habits. The attention paid to the healthfulness of the diet was also associated with a high level of concern regarding the impact of the environment on the food supply in Australian research [10].

Food processing has improved food safety and some aspects of food security by reducing the perishability of some products; however, the over-production, packaging, and consumption of ultra-processed food (UPF) is both harmful to health and the environment, and a reduction in UPF is recommended [11]. Among the Australian population, UPF consumption as a proportion of the total daily energy intake is high. Forty-two percent of the total daily energy intake was from UPF in 2011–2012: the latest Australian National Nutrition and Physical Activity survey [12,13]. Individuals can minimize their impact on the environmental food system by choosing minimally processed and less packaged foods. Specific advice regarding plastics in food packaging, consumer consumption patterns of UPFs, and reducing food waste are not yet fully incorporated into dietary guidelines [1]. Packaging and ingredients, such as sugars, saturated fat, and salt, are added to manufacture UPFs and create consumables that are among the most environmentally unsustainable foods to consume [2]. Public health messaging needs to evolve to reduce the production and consumption of universally accessible and promoted UPFs [14]. 

The Australian National, State, and Local Governments’ waste avoidance strategies have sent clear messages to reduce food and packaging waste. In 2018–2019, households contributed 71% of the reported food waste from the curbside collection in Australia, most of which became landfill [15]. Food production systems contributed to food waste from the paddock to the plate [2,16]. The Australian ‘National Waste Strategy’ sets National Packaging targets to phase out ‘problematic’ food and beverage packaging [15]. The Australian National Food Waste strategy [17] aims to halve food waste by 2030. The strategy aims to intercede at various points along the food system and includes household waste reduction strategies. Household food waste can be reduced through increased awareness of food waste and its impact on the environment [18]. The Australian National Food Waste strategy does not specifically mention educating consumers on choosing less packaged food or measuring attitudes toward food waste. As the actions in the strategy progress, it is important to understand Australian consumer attitudes to potential areas for intervention, for example, the over-packaging of foods and food choices that lead to household food waste. 

There is evidence that consumers are concerned about climate breakdown, viewing it as a global emergency, and that some people are changing their food choices to protect the environment [19]. Concern for changing the environment can encourage environmentally sustainable behaviors that have important consequences for the environment; however, strong social norms or a lack of knowledge can inhibit this change [20]. 

The concept of sustainable diets is relatively new and based on the collaborative efforts of nutrition, medical, and environmental scientists working together to achieve a shared understanding of what constitutes a recommended dietary pattern, for example, the EAT-Lancet Diet. Food-based dietary recommendations vary due to cultural contexts, as does consumer behavior. Much of the existing Australian research explores consumer attitudes regarding one area of sustainable diets, for example, either health or the environment and current consumption. 

Effective advice for sustainable diets needs to be cognizant of consumer attitudes, decision-making processes, and current behaviors. Little is known about the relative attention paid to specific aspects of sustainable diets, for example, the health aspects of diet, levels of food processing, or concern about food waste and packaging disposal. Even less is known about the population subgroups that are most amenable to change. Identifying the demographic characteristics and food decision drivers of those who are already eating healthy, sustainable diets is needed to guide the development of practical, salient dietary advice. This information is important to segment population sub-groups for targeted interventions. 

The objective of this study was to identify the demographic characteristics of Australian adults that pay attention to the healthfulness of their diets, consider the level of food processing when making food decisions, and are concerned about household food waste and packaging disposal. The hypothesis is that there are sociodemographic differences between the population sub-group who pay a lot of attention to these three components of sustainable diets and those that do not.

## 2. Materials and Methods

### 2.1. Study Design and Participants

An online cross-sectional survey of Australian adults (n = 540) over 18 years of age was recruited from an online marketing survey panel. Participants were recruited from an online survey panel by a market research agency https://www.researchify.com.au/ (accessed on 20 December 2022) which was commissioned to conduct the survey. The sample was emailed as an invitation outlining the purpose of the study and a link to the Qualtrics survey containing a downloadable participant information statement and consent form. People with serious illnesses or medical conditions, pregnant women, or anyone who was currently following a special diet for weight loss were excluded. 

### 2.2. Participant’s Attitude to Health, Sustainable Food Waste and Level of Food Processing of Their Diet 

The survey was designed to identify and elicit the individual characteristics of participants, which related to their attitude to healthfulness, food processing, household food waste, and the disposal of food packaging, see Table 1. Survey questions were adapted from the Western Australian Department of Health’s Nutrition Monitoring Survey Series (NMSS) [21]. The questions were developed and assessed by a team of experts (experienced nutrition and dietetics and marketing academics from Curtin University) for content validity. The questions for this study were chosen to enable a comparison with previous studies [21,22,23]. The question on the attention paid to processed food was a direct adaption of the attention paid to the “health aspects of the food I eat” that is routinely used in the NMSS. Household food packaging and food waste questions were adapted from the Australian baseline report on Attitude and Behaviour [8] to include household food waste. Those regarding concern for food waste and packaging were developed from points discussed in Monteiro et al. (2018) and da Silva (2021) [11,22]. 

### 2.3. Data Analysis

Data were extracted from the Qualtrics survey and analyzed using SPSS 28 (IBM SPSS statistics). Responses to the questions related to concern for the healthfulness of food, sustainable food practices concerned with household food waste and food packaging disposal, and the level of processed food were each recoded into a binary outcome: “A lot of attention” Yes or No. Participants who indicated that they “*paid a lot of attention to the health aspect of the food they ate*” were coded as Yes; all other responses were coded as No. Similarly, participants who “*routinely considered the level of processing of the food they ate*” were coded as Yes; all other responses were coded as No. In relation to concern for sustainable food waste and packaging disposal, participants who were *“concerned a lot of the time*” about either: (1) “food packaging I need to dispose of” or (2) “food waste generated in my household” were coded as Yes; all other responses were coded as No. 

Univariate logistic regression analysis was conducted to determine the association of each of the outcomes and the demographic variables, as well as an analysis controlling for age and gender in the model, reported as an odds ratio (OR) with a 95% confidence interval (CI). Binomial multivariate logistic regression was also conducted to determine demographic factors that could identify population groups concerned with healthfulness, sustainable household food, and packaging disposal, as well as the attention paid to the level of processing when making food choices. The categorical demographic factors investigated in the model included gender, age, education level, occupation, income, and body mass index (BMI) (see Table 2 for categories of each variable). 

## 3. Results

### 3.1. Demographic Characteristics of Participants

The demographic characteristics of the participants are presented in Table 2. The participants were generally representative of the Australian population by age, gender, occupation, work pattern, and education level. There was a 12% higher representation in the 55–74-year age group compared to the Australian population data [24]. 

### 3.2. Population Prevalence of Attitudes towards Sustainable Diets

Thirty-seven percent (n = 198) of participants were concerned about household food waste or sustainable food packaging disposal, 28% (n = 153) considered the level of food processing when making their food choice, and 23% (n = 124) paid a lot of attention to the healthfulness of the food they ate. The demographic characteristics and Pearson chi-square results for the binary outcomes are available in the Appendix A. In summary, the income level was significantly different between the positive or negative attention paid to healthfulness (*p* < 0.001) and consideration of food processing (*p* = 0.028); the education level was significantly different for the consideration of food processing (*p* < 0.001) and concern for sustainable food waste and packaging disposal (*p* < 0.001); BMI represented (*p* = 0.023) with concern for sustainable food waste and packaging disposal.

### 3.3. Logistic Regression Analysis to Predict Attention Paid to and Concern for Sustinable Diets 

The logistic regression model analysis for the demographic variables associated with respondents who paid a lot of attention to the health aspects of their diets, considered the level of food processing of their food choices, and were concerned about sustainable household food waste and food packaging disposal, is shown in Table 3A–C. 

#### 3.3.1. Attention Paid to the Health Aspects of Diet by Sociodemographic Factors

Logistic regression analysis found that respondents with a higher income were more likely to pay attention to the healthfulness of their diet (Table 3A). Univariate logistic analysis, controlling for age and gender, found that those with education attainment above Year 12 were one and a half times more likely to pay attention to the health aspects of their diet compared to those of lower educational attainment (OR 1.56 95% CI 1–2.44, *p* = 0.049). 

Respondents earning a higher income, more than AUD$100,000, were twice as likely to pay attention to the health aspects of their diet compared to those earning less (OR 2.06 95% CI 1.32–3.19, *p* = 0.001). Multivariate logistic regression predictive modeling, including all demographic variables, found that the effect of income remained a statistically significant predictor (OR 2.19 95% CI 1.28–3.74, *p* = 0.004). 

#### 3.3.2. Attention to the Intentional Sustainable Food Waste Practices

Thirty-seven percent of participants were concerned about food waste and/or packaging disposal when they made food decisions. Table 3B shows the logistic regression analysis. Univariate analysis found that a higher educational attainment (OR 1.99 95% CI 1.362.97, *p* = <0.001) and annual income (OR 1.52, 95% CI 1.01–2.27, *p* = 0.042) were positively associated with concern for sustainable household food waste and food packaging disposal. Participants who did not work for wages were significantly less likely than those who worked from Monday to Friday (OR 0.57 95% CI 0.38–0.86, *p* = 0.004) to be concerned about food waste and packaging disposal. Multivariate logistic regression analysis modeling showed that those experiencing obesity were half as likely to report concern for food waste and packaging compared to those of a healthy weight (OR 0.55 95% CI 0.32–0.94, *p* = 0.029). Respondents with higher education, above year 12, were 1.9 times as likely to consider sustainable food waste practices compared to the less educated (OR 1.89 95% CI 1.18–3.06, *p* = 0.008). 

#### 3.3.3. Attention Paid to the Level of Processing in the Diet 

Twenty-eight percent of participants indicated concern for the level of food processing when they made food decisions. Table 3C shows the demographic logistic analysis. Univariate analysis, controlling for age and gender, found that those attaining education above year 12 were twice as likely to consider the level of processing compared to those who were less educated (OR 2.18 95% CI 1.42–3.35, *p* < 0.001); participants living in a household with an annual income over AUD$100,000 were twice as likely to pay attention to the processing level of their food compared to those earning less than AUD$50,000 annually (OR 1.88 95% CI 1.23–2.89, *p* = 0.004). Multivariate logistic regression analysis found that the education level remained predictive of consideration for the processing level of food choices (OR 2.04 95% CI 1.22–3.43, *p* = 0.007). 

### 3.4. Attitudes towards Food Healthfulness, Processing Level and Sustainable Food Waste and Packaging Disposal Practices

Table 4 shows the eight groups based on binary outcomes for paying attention to the healthfulness of their diet, consideration of the level of food processing, and concern about sustainable food waste and food packaging. The groupings from the binary outcomes range from those who considered all three aspects (Group 7) to those who did not consider any of the aspects (Group 0) when choosing food. 

The descriptive analyses of the median demographic factors in the highest and lowest levels of attention, consideration, or concern were used to describe the group. The median demographic responses for group 7 (attention to all aspects) were aged 35–44 years, had an annual income of over AUD$100,000, were overweight, attained an education level above Year 12, and worked in professional, managerial, or clerical occupations. The median responses for participants in group 0 (no attention or concern) were similar, with the exception that they were older (median 45–54 years) and earned a lower annual household income of between AUD$50 and 100,000. 

## 4. Discussion

### 4.1. Overall Characteristics by Attitude toward Sustainable Diets 

The current study determined that income and age were the sociodemographic differences between those that paid attention or were concerned about the healthfulness, sustainability, and processing level aspects of their food choices. Those with higher educational attainment were twice as likely to be concerned with sustainable food waste and the processing level of food choices. The majority of participants (45%) did not report a high level of concern for any of the factors, and less than 10% were concerned about the healthfulness, processing level, or sustainable waste practices when making their food choices. Multivariable logistic regression analysis between demographic factors and the potential drivers of food choice found that education level was predictive of considerations for food processing and concern for food waste and food packaging disposal. Income was predictive of the attention paid to the healthfulness of dietary food choices, and the body mass index from self-reported height and weight was negatively associated with concern for sustainable food waste and packaging disposal. 

The findings of this study suggest that there is an opportunity to build awareness about the importance of and need for sustainable diets, specifically related to the increasing concern for food processing levels, the healthfulness of food choices, and sustainable food disposal. Food decisions throughout the day are mostly made subconsciously [25], and influencing individual attitudes, as well as environmental strategies, are important factors that affect dietary change [20]. Despite research suggesting that climate breakdown is viewed as a global emergency by consumers and that some people are changing their food choices for better environmental health [19], this current study suggests that more is needed to change attitudes and behaviors. Increasing both health and environmental literacy related to sustainable diets could support positive action [26]. 

### 4.2. Sustainable Food Waste and Disposal Practices

Over a third of participants were concerned about household food waste and the disposal of food packaging when making their food choices. Australian governments at all levels have introduced support to enhance household recycling practices, and food retailers have made significant changes to reduce the use of plastics and encourage effective recycling [15,17]. These supports for changes in food waste practices and the mass communications concerning them may have influenced attitudes towards disposing of food packaging and food waste. Households contribute a greater proportion of food waste than any other single sector of the food system contributing to the total Australian food waste (34%) of 7.3M tonnes in 2016–2017 [27]. Food manufacturing impacts water, land, and energy use as well as contributing to the greenhouse gas cost incurred with waste disposal [2,11]. A consumer behavior change campaign to reduce food waste has been started in many local government areas across Australia, and estimates suggest that in over 10 years, this could reduce food waste by 1.9M tonnes [17].

The level of education was associated with concern for sustainable food waste and food packaging. This is consistent with previous research that shows better dietary practices among the more educated [28]. To this end, any campaign and support material needs to be clear and simple to understand. There is a policy imperative in Australia to “halve food waste in Australia by 2030” [17]. Almost three-quarters of Australian households report that they do not waste the following types of food: packaged, pantry, dairy, or ready-to-eat/takeaway/take-out or delivered food) [17]. 

The current findings suggest that some consumers may be amenable to simple solution-based advice, encouraging them to buy less food so as not to overeat or waste food and to cook and store food correctly to reduce food spoilage and waste. In the context of the current rising cost of living, the cost-saving benefits of such recommendations may be salient, similar to the successful FoodCent$ program during the economic downturn in the late 1980′s, and abbreviated concepts from the program have been incorporated into contemporary food literacy programs [29,30,31]. The full suite of the original program was based on value for money and health and could be extended to a ‘value for money, health and the environment’ proposition to educate consumers and guide healthy, sustainable food choices. 

Fresh food packaging can extend the life of the product, but the packaging needs to be designed to reduce its contribution to overall household landfill waste. The current findings suggest that specific education is needed to raise consumer awareness of the multiple benefits of sustainable diets and the specific practices to adopt. There is also an opportunity to work with the food retail sector to encourage them to extend their consumer nutrition messages and promotions at the point of sale to incorporate the benefits of sustainable diets and how to achieve them [29].

### 4.3. Processing Level of Food Choices

Twenty-eight percent of participants in the current study considered the level of food processing when choosing food. This is lower than a US study where almost half of the adults paid attention to the level of food processing of the food they purchased [32]. Sample characteristics between the two surveys may account for the differences as the current study included consumers who were not responsible for food purchases, whereas the US study surveyed only household food managers. Food processing has enabled a safer food supply and reduced perishability; however, the long-term health impact of ultra-food processing is causing concern. The strength of evidence suggests that the more processed a food is, the poorer human and planetary health outcomes are increasing [33,34,35,36]. UPFs are displacing nutritious unprocessed and minimally processed foods, and limiting these foods and encouraging minimal processing has been shown to positively impact both diet-related disease risk [37,38] and the environment [3,36]. 

### 4.4. Attention Paid to the Healthfulness of Diet

Less than a quarter (23%) of participants in the current study said that they paid a lot of attention to the health aspects of the food they eat, compared to 16% in a Western Australian study of adults aged 18–30 years in 2015 [8] and 45% of adults over 18 years in 2009 and 2012 [21]. Those who paid attention to the health aspect of their diet were likely to earn a higher income and a higher education level. Diet is known to be positively associated with income and education levels [28,39], and these current study findings are consistent with this. These attitudes influence support for government action to protect the supply of environmentally friendly food; for example, previous Australian research found that people who paid a lot of attention to the health aspects of the food believed in the importance of government control and regulatory measures to protect the supply of environmentally friendly food [23].

As the level of food processing was a consideration for 37% of participants in the current study, information about the impact of food processing on human and planetary health may be of interest to these consumers. Communicating the findings of recent research regarding UPF consumption on health and the environment to this demographic is recommended, and these communications may increase concern among those who are currently unaware of their impact. 

### 4.5. Strengths and Limitations

The participants in this study were similar to the Australian population based on age, gender, and occupation when compared to the 2021 census [16]. A strength of this study is that the inclusive criteria enabled a wider perspective of attitudes as it included both household food managers and other household members. Consumer food decisions in the home are part of the food system and within an individual’s immediate control. All members of a household have an opportunity to consider food and packaging waste, regardless of how engaged they are in the food management of their household. Household food management includes food planning, procurement, preparation, serving, and cleaning up [24]. It is, therefore, important to consider all members of the household when developing messages to encourage the wider population to choose a healthier, sustainable, and minimally processed diet.

The sample size (n = 540) is a small representation of the Australian population recruited through a marketing survey panel. A possible limitation is that the survey panelists are members of a marketing panel rather than a random sample of Australians and may differ from the population. The authors expected a wider range of demographic factors to impact the food choice drivers being investigated. The participants that were recruited represented a slightly higher proportion of Australians over 55 years compared to the Australian population data [16]. Aging populations may be more likely to have a chronic disease, such as heart disease, type 2 diabetes, or obesity, as these increase with age [34], and this may impact the attention paid to the diet. 

### 4.6. Generalizability

This study is unique as it has looked at the combined effect of three synergistic ecological aspects of food decisions, namely healthfulness, the level of food processing, and sustainable household food waste and food packaging disposal. The results may not be generalizable to other countries, as attitudes to diet and environmental sustainability may vary. 

## 5. Conclusions

Engaging with consumers to elicit the key drivers of their food choices is critical for the development of salient dietary advice. Just under half of the Australian adults surveyed in this study did not prioritise the health, processing level, and food or packaging waste when making food decisions. Interventions to raise consumer awareness of the importance of sustainable diets and the specific strategies to achieve them are warranted. Understanding food decisions related to sustainable diets in specific population groups is critical to reinforce and supporting public health action. 

## Figures and Tables

**Figure 1 ijerph-20-02633-f001:**
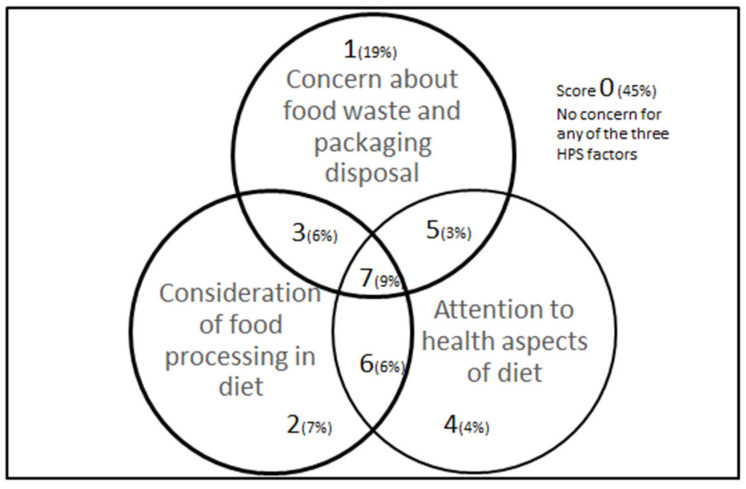
Venn diagram of the healthfulness, processing level, and sustainable food waste and packaging disposal attitudes of participants.

**Table 1 ijerph-20-02633-t001:** Healthy Eating Message Design (HEMD) Survey questions measuring attention paid to health aspects, the level of food processing of a participant’s diet, and concern about food waste and disposal of food packaging.

Factor (Source)	Question
**Demographics**	
	Please confirm your gender. How would you best describe yourself with respect to ethnicity? What is your age? What is your body weight/height? What is the highest level of education you have completed? What is your annual income before tax? What best describes your current work pattern? What is your occupation?
**Attention paid to health aspects of diet** [21]	
1 = Pay a lot of attention. 2–5 = Do not pay a lot of attention	Which statement best describes how your feel about your diet? I pay a lot of attention to the health aspect of the food I eat to make sure my diet is as healthy as possible. I take a bit of notice of the health aspect of the food I eat to make sure I have a fairly good diet. I don’t really think much about the health aspect of the food I eat. I don’t think at all about the health aspect of the food I eat. Unsure/don’t know.
**Consider processed foods adapted from the question above**	
1 = Pay a lot of attention 2–5 = Do not pay a lot of attention	Which statement best describes how you think about processed foods in your diet? I routinely consider how processed the food I eat is. I take a bit of notice of how processed the food I eat is. I don’t really think much about how processed the food I eat is. I don’t think at all about how processed the food I eat is. Unsure/don’t know.
**Concern about sustainability aspects of food choices** [11,22]	
3 = Concerned 1, 2 = Not concerned	How concerned are you about these aspects of your food choice? Food packaging I need to dispose of: 1. Do not consider; 2. Some of the time; 3. A lot of the time. Food waste generated in my household: 1. Do not consider; 2. Some of the time, 3. A lot of the time

**Table 2 ijerph-20-02633-t002:** Demographic characteristics of participants compared to the 2019b Australian census data.

	N (%)	ABS Census 2021 (%) [16]
Gender (n = 538)		
Male	271 (50.2%)	49.3%
Female	263 (48.7%)	50.7%
Non binary	4 (0.7%)	0.17%
Age (years) (n = 539)		
18–24	55 (10.2%)	6.2% (20–24)
25–34	86 (16.0%)	14.3%
35–44	94 (17.4%)	13.7%
45–54	84 (15.6%)	12.8%
55–64	99 (18.4%)	11.8%
65–74	82 (15.2%)	9.7%
Over 75	39 (7.2%)	7.5%
Ethnicity		
White	455 (84.3%)	
Aboriginal and Torres Strait Islander	17 (3.1%)	3.2%
Asian	57 (10.6%)	
Other	18 (3.4%)	
Educational attainment		
Year 10 or below	75 (13.9%)	10.8% *
Year 12 or equivalent	112 (20.7%)	15.7% *
Trade/apprenticeship	34 (6.3%)	Certificate III, IV 15.7% *
Advanced diploma or certificate	151 (28.0%)	30%
University Bachelor’s degree or higher	164 (30.4%)	31.2%
Unsure/don’t know	4 (0.7%)	
Annual income (AUD) **		
Below $25,000	120 (22.2%)	
$25,000–$49,999	130 (24.1%)	
$50,000–$74,999	104 (19.3%)	
$75,000–$99,999	75 (13.9%)	
Over $100,000	81 (15.0%)	
Prefer not to say	30 (5.6%)	
Work pattern (n = 535)		
Mon-Fri, 9–5	216 (40%)	
Shift worker	43 (8.0%)	
Fly-in Fly-out ^#^	6 (1.1%)	
Not working for wages	199 (36.9%)	Unemployed 6.9% Away 5.0% *
Other	71 (13.1%)	
Part-time or casual		30.4% *
Occupation (n = 425)		
Professional	94 (20.9%)	22.2% *
Manager	60 (13.3%)	13% *
Clerical or Administration	96 (17.8%)	13.6% *
Laborer and Technical or Trade	87 (19.3%)	23% *
Retired and Homemaker	88 (19.6%)	
BMI (m/kg^2^) (n = 490)		
<18.0	15 (2.8%)	
18–24.9	179 (33.1%)	
25–29.9	154 (28.5%)	
30–34.9	69 (12.8%)	
>35	73 (13.5%)	

* Data from 2016 Australian Bureau of Statistics census [17]. ^#^ fly-in fly-out refers to workers who are flown into their worksite for the duration of their rosters, before flying home. Often in the mining, construction, oil, and gas industries. ** Median weekly household gross income 2019–2020 $1, 786 ($92K annual).

**Table 3 ijerph-20-02633-t003:** (**A**): Logistic regression for association between sociodemographic characteristics and ‘attention paid’ to the health aspects of their diets (n= 540). (**B**) Logistic regression for association between sociodemographic characteristics and the likelihood of being concerned for sustainable household food waste and food packaging disposal (n = 540). (**C**) Logistic regression for association between socio-demographic characteristics and likelihood of considering the level of food processing (n = 540).

Variable		Univariate OR (95% CI, *p* Value)	Univariate Adjusted for Age, Gender OR (95% CI, *p* Value)	Multivariable OR (95% CI, *p* Value)
(**A**)				
Gender	Male	1		
	Female	0.82 (0.55–1.23, *p* = 0.347)		
Age group	18–34 years	1		
	35–54 years.	1.34 (0.78–2.31, *p* = 0.284)		
	55 years and over	1.37 (0.82–2.31, *p* = 0.232)		
BMI	Healthy weight	1	1	
	Overweight	0.79 (0.47–1.31, *p* = 0.355)	0.69 (0.41–1.17, *p* = 0.175)	
	Obese	0.704 (0.414–1.19, *p* = 0.196)	0.64 (0.37–1.10, *p* = 0.110)	
Education	Year 12	1	1	
	Beyond year 12 (Trade/tertiary)	1.52 (0.97–2.36, *p* = 0.065)	1.56 (1.00–2.44, *p* = 0.049)	
Household Income (AUD)	<$50K	1	1	1
	$50–100K	0.84 (0.46–1.55, *p* = 0.585)	0.89 (0.48–1.66, *p* = 0.731)	1.06 (0.53–2.12, *p* = 0.860)
	>100K	**2.06 (1.32–3.19, *p* = 0.001)**	**2.17 (1.38–3.41, *p* < 0.001)**	**2.19 (1.28–3.74, *p* = 0.004)**
Occupation *	Not Office worker	1	1	
	Office	1.16 (0.74–1.82, *p* = 0.511)	1.23 (0.78–1.95, *p* = 0.371)	
Employment	Monday-Friday	1	1	
	Shift work	0.59 (0.25–1.42, *p* = 0.245)	0.64 (0.27–1.53, *p* = 0.32)	
	FIFO #	0	0	
	Not for wages	0.93 (0.60–1.43, *p* = 0.735)	0.86 (0.54–1.37, *p* = 0.533)	
	Part time casual	1.19 (0.47–3.02, *p* = 0.705)	1.11 (0.43–2.85, *p* = 0.821)	
(**B**)				
Gender	Male	1		
	Female	1.31 (0.92–1.87, *p* = 0.129)		
Age group	18–34 years	1		
	35–54 years.	0.93 (0.59–4.46, *p* = 0.763)		
	55 years and over	0.72 (0.46–1.11, *p* = 0.137)		
BMI	Healthy weight	1	1	1
	Overweight	1.31 (0.85–2.02, *p* = 0.218)	1.44 (0.92–2.25, *p* = 0.111)	1.57 (0.98–2.54, *p* = 0.0062)
	Obese	0.67 (0.42–1.07, *p* = 0.092)	0.69 (0.43–1.13, *p* = 0.146)	**0.55 (0.32–0.94, *p* = 0.029)**
Education	Year 12	1	1	1
	Beyond year 12 (Trade/tertiary)	**1.99 (1.35–2.94, *p* < 0.001)**	**2.01 (1.36–2.97, *p* < 0.001)**	**1.89 (1.18–3.06, *p* = 0.008)**
Household Income (AUD)	<$50K	1	1	
	$50–100K	1.18 (0.73–1.90, *p* = 0.499)	1.17 (0.72–1.91, *p* = 0.516)	
	>100K	1.48 (0.99–2.19, *p* = 0.051)	**1.52 (1.01–2.27, *p* = 0.042)**	
Occupation *	Not Office worker	1	1	
	Office	1.13 (0.77–1.65, *p* = 0.542)	1.06 (0.72–1.58, *p* = 0.762)	
Employment	Monday-Friday	1	1	
	Shift work	0.99 (0.51–1.92, *p* = 0.974)	0.99 (0.51–1.95, *p* = 0.980)	
	FIFO #	0.67 (0.12–3.83, *p* = 0.668)	0.71 1(0.13–3.99, *p* = 0.698)	
	Not for wages	**0.57 (0.38–0.84, *p* = 0.004)**	**0.57 (0.38–0.86, *p* = 0.008)**	
	Part time casual	1.08 (0.47–2.49, *p* = 0.858)	1.08 (0.46–2.52)	
(**C**)				
Gender	Male	1		
	Female	0.89 (0.62–1.31, *p* = 0.580)		
Age group	18–34 years	1		
	35–54 years.	1.63 (0.98–2.71, *p* = 0.060)		
	55 years and over	1.52 (0.93–2.49, *p* = 0.095)		
BMI	Healthy weight	1	1	
	Overweight	1.18 (0.74–1.89, *p* = 0.482)	1.09 (0.67–1.76, *p* = 0.735)	
	Obese	0.85 (0.51–1.39, *p* = 0.517)	0.78 (0.47–1.31, *p* = 0.352)	
Education	Year 12	1	1	1
	> Year 12 (Trade/tertiary)	**2.06 (1.35–3.16, *p* < 0.001)**	**2.18 (1.42–3.35, *p* < 0.001)**	**2.04 (1.22–3.43, *p* = 0.007)**
Household Income (AUD)	<$50K	1	1	
	$50–100K	1.03 (0.61–1.75, *p* = 0.905)	1.12 (0.66–1.92, *p* = 0.667)	
	>100K	**1.70 (1.12–2.58, *p* = 0.012)**	**1.88 (1.23–2.89, *p* = 0.004)**	
Occupation *	Not Office worker	1	1	
	Office	1.03 (0.69–1.56, *p* = 0.871)	1.07 (0.71–1.64, *p* = 0.730)	
Employment	Monday-Friday	1	1	
	Shift work	0.82 (0.39–1.72, *p* = 0.593)	0.87 (0.41–1.84, *p* = 0.713)	
	FIFO #	0.47 (0.05–4.15, *p* = 0.501)	0.51 (0.05–4.4, *p* = 0.538)	
	Not for wages	0.82 (0.54–1.25, *p* = 0.359)	0.721 (0.46–1.12, *p* = 0.145)	
	Part time casual	2.19 (0.95–5.06, *p* = 0.066)	1.93 (0.87–4.52, *p* = 0.128)	

* Occupation was grouped into office based: professional, managerial, or clerical occupation and not office based: laboring occupations, retired, or a homemaker. # FIFO fly in fly out refers to workers who are flown into their worksite for the duration of their rosters, before flying home. Often in the mining, construction, oil, and gas industries. Yes group: univariate analysis, univariate after adjusting for age and gender and multivariate analysis.

**Table 4 ijerph-20-02633-t004:** Population groupings based on attitude towards health, processing, and sustainable food waste and packaging disposal (HPS), n = 540 Australian adults 18 years and over.

Pay Attention to the Health Aspects in Diet	Consider the Level of Food Processing in Diet	Concern for Sustainable Food Waste and Disposal	HPS * Group	Proportion (%)
Yes	Yes	Yes	7	8.9
Yes	Yes	No	6	6.3
Yes	No	Yes	5	3.3
No	Yes	Yes	4	5.7
Yes	No	No	3	4.4
No	Yes	No	2	7.4
No	No	Yes	1	18.7
No	No	No	0	45.2

* HPS health, processing and sustainable food waste and packaging disposal. Forty-five percent of participants did not show a high level of attention or concern for any of the three factors, 19% were concerned about sustainable household packaging disposal and food waste, and less than 10% considered all three aspects when choosing their diet, see Figure 1.

## Data Availability

Data are available on request from the corresponding authors.

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
