# Peer review of "Australian Consumers’ Attitudes towards Sustainable Diet Practices Regarding Food Waste, Food Processing, and the Health Aspects of Diet: A Cross Sectional Survey"

_ijerph, 2023, doi:10.3390/ijerph20032633_

Round 1

Reviewer 1 Report

Overall comment

Throughout the text formatting is in order. Example, line 235.

References in line 237 must be corrected.

Reference Section

All references must be correctly presented, according to the journal referential.

Reference [28] dates from 1998, it should be updated or justified.

Author Response

We would like to thank the reviewers for their valuable feedback and the comments that overall the topic of this research was interesting. A detailed response to the reviewers’ comments is outlined below.

Line

Referee

Comment

Response

References

1

Throughout the text formatting is in order. Example, line 23

References have been edited throughout.

Section 3.4

1

References in line 237 must be corrected.

References have been edited throughout.

References

1

All references must be correctly presented, according to the journal referential

References have been edited throughout.

Section 4.4

1

Reference [28] dates from 1998, it should be updated or justified.

Edited Section 4.4 to justify the 1998 reference: … line 335 includes, “… and abbreviated concepts from the program have been incorporated into contemporary food literacy programs [28-30]. The full suite of the original program was based on value for money and health, and, could be extended to a value for money, health and the environment proposition to educate consumers and guide healthy sustainable food choices.”

Reviewer 2 Report

Interesting work. I think that in the discussion, the measures regarding the future actions envisaging the education of consumers should be better highlighted. Please find attached some minor suggestions.

Author Response

We would like to thank the reviewers for their valuable feedback and the comments that overall the topic of this research was interesting. A detailed response to the reviewers’ comments is outlined below. 

Line

Referee

Comment

Response

Line 54

2

entite

Edited. Entitled

Line 85

2

waste,

Edited. Changed to full-stop

Table 1

2

[21])-

Deleted )-

Table 1

2

I don’t really think much about how processed of the food I eat

I don’t think at all about how processed of the food I eat

Edited

Table 1

2

Delete )

Done

Table 2

2

Added space before (

Done

Results

2

Line 170-172 consistent spacing p values

Done

Results

2

Line 173 delete space

Done

Results

2

Line 178 insert line

Done

Results

2

Line 180 delete.

Done

Table 3

2

Consistent space before %

Done

Section 3.3.2

2

Line 197, 8 Revise sentence

Done

2

Line 201, correct p value

Done

2

Line 208 insert line

Done

Table 3

2

Consistent space before %

Done

2

Line 216 add line

Done

Table 3C

2

Consistent space before %

Done

2

Line 230-237 edit text

Done

Table 4

2

Line 243 explain HPS

Done

Section 4.3

2

Line 291, 293 Verify spacing

Done

2

Line 304, clarify sentence

Done

2

Line 316, delete bracket

Done

2

Line 329 define WA

Done

Discussion-Section 4.3, third paragraph

2

I think in the discussion, the measures regarding the future actions envisaging the education of consumers should be better highlighted.

We have added more detail regarding future actions and specific communications for consumers and the potential channels, e.g. point of sale. Line 335

“…, and abbreviated concepts from the program have been incorporated into contemporary food literacy programs [28-30]. The full suite of the original program was based on value for money and health, and, could be extended to a ‘value for money, health and the environment’ proposition to educate consumers and guide healthy sustainable food choices.

Fresh food packaging can extend the life of the product, but the packaging needs to be designed to reduce its contribution to overall household landfill waste. The current find-ings suggest that specific education is needed to raise consumer awareness of the multiple benefits of sustainable diets and specific practices to adopt. There is also an opportunity to work with the food retail sector to encourage them to extend their consumer nutrition messages and promotions at point-of-sale to incorporate the benefits of sustainable diets and how to achieve them [28].’

Reviewer 3 Report

The main objective of this study is about to identify the demographic characteristics of Australian adults based on the level of attention they paid to the healthfulness of their diet, consideration of the level of food processing, and concern about household food waste and sustainable packaging disposal. The topic is interesting, however, need some improvement.

1.     In introduction, I can see just previous studies and the objective of the study. I haven’t seen any gaps from previous studies.

2.     From the beginning I can answer the research questions, so please bring more evidence why we need such research?

3.     In the last paragraph of the introduction the authors mentioned that “The objective of

4.     this study was to identify the demographic characteristics of Australian adults who pay attention to the healthfulness of their diets, consider the level of food processing when making food decision, and are concerned about household food waste and packaging disposal.”

It doesn’t look like an objective. It is more toward an output. Please rewrite it.

5.     What is the novelty of this study?

6.     It is better the authors come up with research hypotheses section and prepare every hypothesis based on their background of the study.

7.     Sample size calculation is not clear. How did the authors find that 540 participants is enough? Did they do pilot study? How about validity and reliability of their questionnaire?

8.     Line 92, the authors mentioned “The Australian strategy does not mention educating consumers on choosing less packaged food or measuring attitudes toward food waste.” It’s not clear to me about Australian strategy. What do they mean the Australian strategy?

9.     Line 94, “Australian consumer attitudes to over packaging of foods and food choices that lead to waste is limited” need to rewrite and is it from the authors or other study?

10.  Section 2, how did the authors approach to the participants? Where did they receive their contacts?

11.  Table 2, total number of male, female, and non binary is not equal 540. It can be see in other research variables in Table 2 like ethnicity, work pattern. Please check all them.

12.  Line 237 please rewrite it “Error! Reference source not found.4”

Author Response

Journal IJERPH (ISSN 1660-4601)

Manuscript ID ijerph-2140864-Repsonse to reviewer comments.

General comments: We would like to thank the reviewers for their valuable feedback and the comments that overall the topic of this research was interesting. A detailed response to the reviewers’ comments is outlined below.

Specific line comments and edits are described in the attached document.

Round 2

Reviewer 3 Report

The Authors amended all of my comments. The manuscript has enough quality to publish. Thank you!